# Power Capability Analysis of Lithium Battery and Supercapacitor by Pulse Duration

**Seongjun Lee [1] and Jonghoon Kim [2],***

[1] School of Mechanical System & Automotive Engineering, Chosun University, Gwangju, 61452, Korea; lsj@chosun.ac.kr
[2] Department of Electrical Engineering, Chungnam National University, Daejeon, 34134, Korea
* Correspondence: whdgns0422@cnu.ac.kr; Tel.: +82-42-821-5657

**Abstract:** In this report, a method for estimating pulse power performance according to pulse duration is proposed. This approach can be used for power control logic in an environmentally friendly power generation system such as electric vehicles and an energy storage system (ESS). Although there have been studies on pulse power capability, we are unaware of any publications on the estimation of the magnitude of pulse power according to the power usage time, and the verification of the estimation result. Therefore, we propose a method to predict power performance according to the pulse duration of batteries and supercapacitors that are used in eco-friendly power generation systems. The proposed method is systematically presented using both a lithium-ion battery module with a nominal voltage of 44 V, 11 Ah, and a supercapacitor module with a maximum voltage of 36 V and a capacitance of 30 F.

**Keywords:** pulse power capability; energy storage system; battery; electric vehicle; supercapacitor

---

## 1. Introduction

Generally, the pulse power capability, a term that is commonly used in energy storage systems such as batteries and supercapacitors, is the maximum output power that does not exceed the over- and under-voltage limit conditions in the current state of charge (SOC) of the energy storage. Therefore, even in the case of batteries and supercapacitors that are used in hybrid cars, battery, and hydrogen-electric vehicles, the pulse power capability is a very important parameter in terms of electrical stability as well as driving performance [1–5]. The relationship between pulse power and the driving performance and safety of an electric vehicle is represented by the block diagram shown in Figure 1.

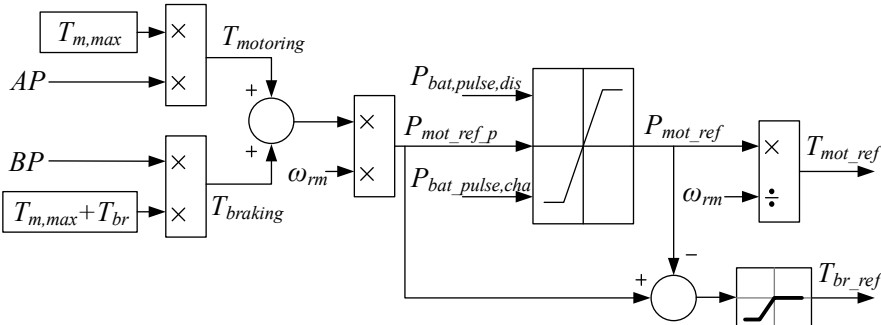

**Figure 1.** Block diagram for the generation of traction and braking torque reference of an electric vehicle.

Figure 1 shows a basic control block used to generate a torque command for a traction motor and the mechanical brake which is implemented in the vehicle control unit (VCU) of an electric vehicle.

The VCU determines the torque command, denoted $T_{mot\_ref}$, that allows the battery to be driven by the power it can supply without operating at a low voltage, based on the motor speed, denoted $\omega_{rm}$, as the vehicle's accelerator pedal (AP) is pressed. Similarly, in the braking mode, the VCU calculates the regenerative torque command of the traction motor and the torque command value of the mechanical brake. This is performed so that the traction motor can be operated within the maximum chargeable power value without exceeding the overvoltage of the battery, based on the degree of depression of the brake pedal (BP). The available power of the battery for determining the torque command can be defined as the pulse power, which is represented as the variables $P_{bat,pulse,dis}$ and $P_{bat,pulse,cha}$ in Figure 1. Therefore, to maximize the driving performance of a vehicle and to stabilize the system, it is necessary to estimate the maximum pulse power that energy storage devices such as batteries and supercapacitors can charge and discharge in the current state.

Studies on the pulse power capability of auxiliary storage devices and their estimation have been conducted [6–11]. In a previous paper [6], a method for calculating the pulse power and an experimental approach for measuring the lumped resistance were described. However, this work did not consider an approach for vehicle driving control and the effect on the duration time when pulse power is applied. In related studies [7,8], the magnitude of the pulse power was estimated over a period of 10 seconds, considering the acceleration and deceleration of the electric vehicle. Bjorn Fridholm [9] proposed an adaptive power capability estimation method that considered the communication time delay between controllers. This study focused mainly on stability analysis of the feedback system with batteries. Rui X. [10] presented a recursive least square method for estimating peak power capability based on the dynamic battery model. However, the effects of the pulse duration were not fully considered. In addition, there are no known studies on the supercapacitors that are typically used to assist the peak power. Studies on the pulse power of a supercapacitor over a 5 second or 10 second duration are presented in references [11,12]. However, the magnitude of the lumped resistance measured based on the experimental results for the hybrid pulse power characterization (HPPC) test [6] was used, and there was no consideration of the change of the pulse duration.

Therefore, in this investigation, we propose a method to estimate the pulse power of a battery and supercapacitor according to the sampling period of the power control algorithm of each application. Firstly, a method for estimating the magnitude of the lumped resistance according to the pulse duration is presented to determine the pulse power of a battery and a supercapacitor. It is subsequently shown that a battery and supercapacitor can be used in the stable voltage range for pulse power operation when the proposed method is applied. In this paper, pulse power estimation results for a 44.4 V, 11 Ah lithium battery module, and a 36 Vmax, 30 F supercapacitor module are presented to demonstrate the validity of the proposed method.

## 2. Pulse Power Capability of a Lithium Battery

### 2.1. Lumped Resistance of a Battery

In Figure 2, when a pulse current of I [A] is applied to the battery for a period $t_1$–$t_2$, the values $V_{bat,t1}$ and $V_{bat,t2}$ can be measured from the voltage response of the battery. Based on this experimental result, it is possible to obtain the lumped resistance, which is the impedance of the battery, by applying the pulse current to the battery and using Equation (1), i.e., Ohm's law [7,8,13,14].

$$R_{bat,lumped} = \frac{\Delta V}{\Delta I} = \frac{V_{bat,t1} - V_{bat,t2}}{I} \tag{1}$$

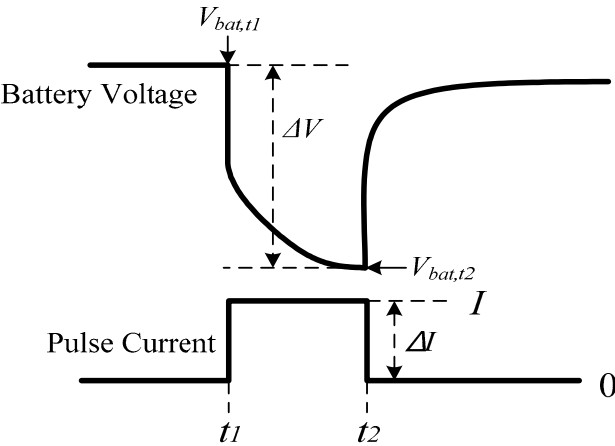

**Figure 2.** Battery voltage response during the pulse current.

However, since that the lumped resistance of the battery is a function of the time when the constant current is applied, as can be seen from Equation (1), it should be calculated according to the period for which the pulse power is required. As indicated in a previous work [6], hybrid and electric vehicles typically use a lumped resistance measured at a 10 second duration to calculate the pulse power. However, to maximize battery utilization, the lumped resistances must be recalculated as follows [7].

To calculate the lumped resistance, it is necessary to determine the voltage ($V_{bat, t1}$) at the time when the pulse current is applied and the battery voltage ($V_{bat, t2}$) at the time when the pulse current ends. Given that $V_{bat, t1}$ is the voltage measured at the present time and $V_{bat,t2}$ is the voltage at a future time, the latter must be predicted to calculate the lumped resistance in real-time. Therefore, in this paper, $V_{bat,t2}$ is predicted using an electrical equivalent model of the battery as shown in Figure 3.

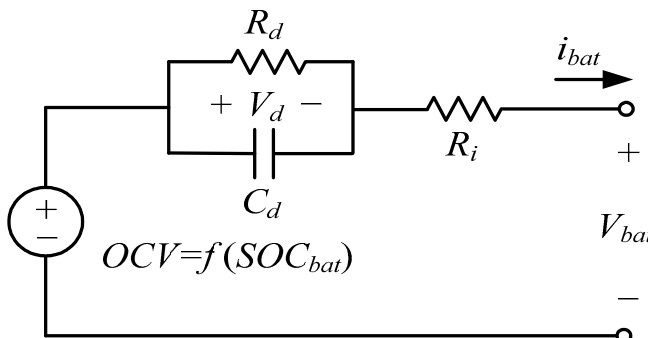

**Figure 3.** Electrical equivalent circuit model of lithium battery.

Assuming that a pulse current is applied at time $t_1$, the voltage of the battery can be expressed by Equation (2). In this case, the decrease of the battery SOC during the pulse current injection time can be expressed by Equation (3) and the voltage $V_d$ across the capacitor $C_d$ at the time $t_2$ can be expressed by Equation (4). Therefore, given that the voltage of the battery at time $t_2$ is derived as expressed by Equation (5), the lumped resistance of the battery can be calculated using the predicted voltages $V_{bat,t2}$ and Equation (1).

$$V_{bat,t1} = OCV\left(SOC_{bat,t1}\right) - V_{d,t1} \tag{2}$$

$$SOC_{bat,t2} = SOC_{bat,t1} - \frac{(t_2 - t_1)}{C_n}I \tag{3}$$

$$V_{d,t2} = e^{-\frac{1}{R_dC_d}(t_2-t_1)}V_{d,t1} + R_dI\left(1 - e^{-\frac{1}{R_dC_d}(t_2-t_1)}\right) \tag{4}$$

$$V_{bat,t2} = OCV\left(SOC_{bat,t2}\right) - V_{d,t2} - R_iI \tag{5}$$

## 2.2. Pulse Power Capability of a Battery

If the battery's lumped resistance is known, it is possible to estimate the magnitude of the maximum pulse current for which an under-voltage or over-voltage does not occur in any charged status of the battery. The maximum pulse current follows the definition of the lumped resistance and is derived as Equations (6) and (7). However, if the predicted pulse current exceeds the maximum current specification of the cell provided by the manufacturer, or exceeds the current specification of the power converter or the overcurrent protection relay, the maximum pulse current is limited according to Equations (8) and (9).

$$I_{bat,pulse,dis} = \frac{V_{bat,t1} - V_{bat,\min}}{R_{bat,lumped,dis}} \tag{6}$$

$$I_{bat,pulse,cha} = \frac{V_{t1} - V_{bat,\max}}{R_{bat,lumped,cha}} \tag{7}$$

$$I_{bat,pulse,dis} = \min\left\{I_{bat,pulse,dis}, I_{bat,dis,\max,set}\right\} \tag{8}$$

$$I_{bat,pulse,cha} = \max\left\{I_{bat,pulse,cha}, I_{bat,cha,\max,set}\right\} \tag{9}$$

In this case, $V_{bat\_min}$ represents the low-voltage limit value indicated by the battery manufacturer or the low-voltage limit value set for the system operation, whereas $V_{bat\_max}$ corresponds to the over-voltage limit value stipulated by the manufacturer or the over-voltage limit value set for the system operation. $R_{bat,lumped,dis}$ and $R_{bat,lumped,cha}$ represent lumped resistances for discharging and charging, respectively. $I_{bat,dis,max,set}$ and $I_{bat,cha,max,set}$ represent the current limit value determined by the smallest value among the maximum current values of the power converter and the battery cell.

Therefore, the maximum pulse power that can be charged and discharged in the current status of the battery is defined as Equations (10) and (11), which represents the constant power capability that can provide or store the energy to the load for a specified time. As such, when the pulse power information of the battery is used for power control, the battery can be used within a stable voltage range, and a system control algorithm can be implemented to improve the performance and efficiency of the vehicle.

$$P_{bat,pulse,dis} = \begin{cases} OCV \cdot I_{bat,dis,\max,set} - I^2_{bat,dis,\max,set} \cdot R_{bat,lumped,dis} & if \ \ I_{bat,pulse,dis} \geq I_{bat,dis,\max,set} \\ \frac{V_{bat,\min}\left(V_{bat,t1} - V_{bat,\min}\right)}{R_{bat,lumped,dis}} & if \ \ I_{bat,pulse,dis} < I_{bat,dis,\max,set} \end{cases} \tag{10}$$

$$P_{bat,pulse,cha} = \begin{cases} OCV \cdot I_{bat,cha,\max,set} + I^2_{bat,cha,\max,set} \cdot R_{bat,lumped,cha} & if \ \ I_{bat,pulse,cha} \geq I_{bat,cha,\max,set} \\ \frac{V_{bat,\max}\left(V_{bat,t1} - V_{bat,\max}\right)}{R_{bat,lumped,cha}} & if \ \ I_{bat,pulse,cha} < I_{bat,cha,\max,set} \end{cases} \tag{11}$$

## 3. Pulse Power Capability of a Supercapacitor

### 3.1. Lumped Resistance of a Supercapacitor

The pulse power capability of a supercapacitor has been investigated using the same approach used for a battery [12]. In this report, we first examine the pulse power capability of supercapacitors using the same concept of pulse power that was defined for a battery. However, the use of pulse power values based on the conventional definition used in the power control algorithm leads to the problem whereby the output performance values differ slightly from the available maximum pulse power. Therefore, in the case of a supercapacitor, we propose a method for calculating the lumped resistance and the pulse power according to the sampling period, in which the pulse power is calculated and this information is utilized in the power control algorithm.

If a pulse current is applied to a supercapacitor, the response of the voltage and current can be obtained as shown in Figure 4. Thus, the supercapacitor can be modeled as the electrical equivalent circuit shown in Figure 5. At this time, the magnitude of the lumped resistance can be obtained from Equation (12).

$$R_{sc,lumped} = \frac{\Delta V_{sc}}{\Delta I_{sc}} = \frac{V_{sc,t1} - V_{sc,t2}}{I_{sc}} \tag{12}$$

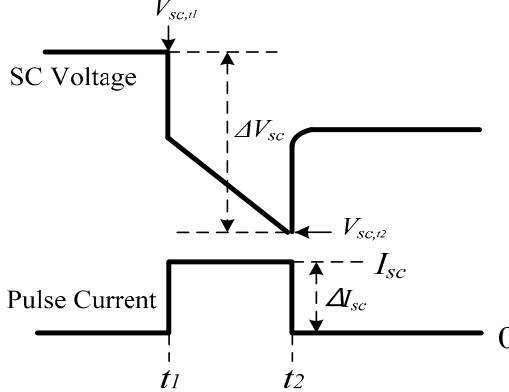

**Figure 4.** Voltage response of supercapacitor during the pulse current.

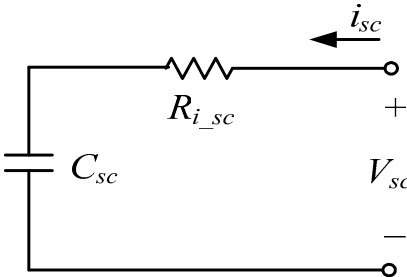

**Figure 5.** The equivalent electrical circuit of the supercapacitor.

In this case, if the duration of the pulse power changes, the lumped resistance should be calculated according to the time. As indicated earlier, given that a supercapacitor can be modeled as an electrical equivalent circuit as shown in Figure 5, the voltage after the sampling period can be estimated as shown in Equation (13). Thus, using the current and predicted voltages, the lumped resistance value can be expressed by Equation (14).

$$
\begin{aligned}
V_{sc,t2} &= V_{sc,t1} - R_{i,sc}I_{sc} - \int_0^{t_2-t_1} \frac{I_{sc}}{C_{sc}} dt \\
&= V_{sc,t1} - R_{i,sc}I_{sc} - \frac{\Delta T I_{sc}}{C_{sc}} = V_{sc,t1} - \left(R_{i,sc} + \frac{\Delta T}{C_{sc}}\right)I_{sc}
\end{aligned}
\tag{13}
$$

$$R_{sc,lumped} = R_{i,sc} + \frac{\Delta T}{C_{sc}} \tag{14}$$

In this case, $V_{sc,t1}$ is the voltage at $t_1$, $V_{sc,t2}$ is the voltage at $t_2$ when the pulse current ends, $R_{i,sc}$ is the equivalent series resistance of the supercapacitor, $C_{sc}$ is the capacitance of the supercapacitor, $I_{sc}$ is the magnitude of the pulse current, $\Delta T$ is the time duration of the pulse current and is given by $\Delta T = t_2 - t_1$.

### 3.2. Pulse Power Capability of a Supercapacitor

To obtain the maximum available discharge and charge pulse power within a set voltage range of a supercapacitor, S.M. Lukic [12] used Equations (15) and (16) to define the battery's pulse power.

$$P_{sc,pulse,dis} = \frac{V_{sc,\min}(V_{sc,t1} - V_{sc,\min})}{R_{sc,lumped}} \tag{15}$$

$$P_{sc,pulse,cha} = \frac{V_{sc,\max}(V_{sc,t1} - V_{sc,\max})}{R_{sc,lumped}} \tag{16}$$

In this case, $V_{sc,max}$ is the maximum voltage of the supercapacitor, and $V_{sc,min}$ is the low-voltage limit value, which is usually set to half the maximum voltage ($0.5 \times V_{sc,max}$). The parameter $R_{sc,lumped}$ represents the lumped resistance of the supercapacitor, defined as Equation (14).

## 4. Experimental Results

The charge and discharge tests, and the pulse power verification of the battery and supercapacitor were performed using the experimental setup shown in Figure 6. Figure 6a represents the equipment used for the charge and discharge experiments to model the battery's cells, whereas Figure 6b shows the experimental configuration of the battery and supercapacitor modules. Table 1 lists the batteries, and supercapacitors used in the experiment, in addition to equipment used for charging and discharging.

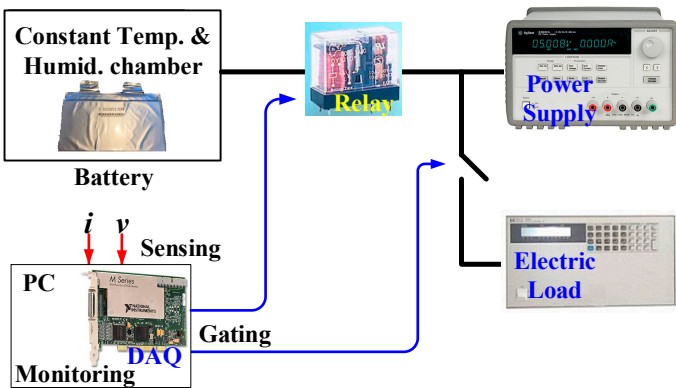

(**a**) Experimental setup for a battery cell

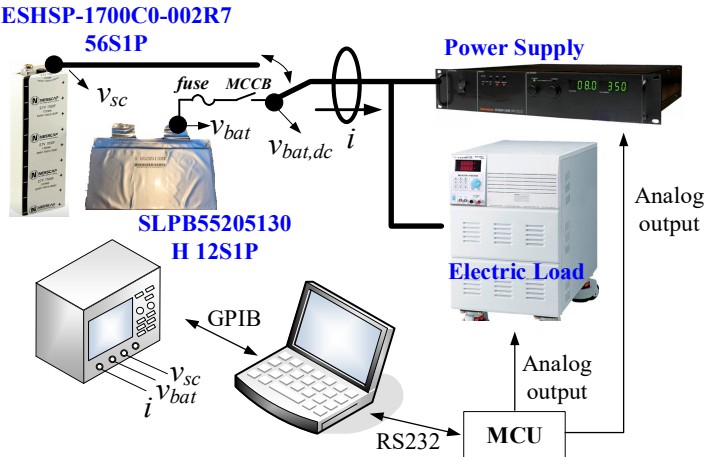

(**b**) Experimental setup for the battery and supercapacitor module

**Figure 6.** Experimental configuration of the battery and supercapacitor.

To obtain the pulse power of the battery as described in Section 2, it is necessary to measure or estimate the lumped resistance of the battery. An example of an experimental method for measuring this resistance is the direct current internal resistance (DCIR) method, similar to the current profile and experimental procedure in Figure 7 [15]. In this method, when the discharge (denoted using a positive sign) and charge current (denoted using a negative sign) are increased stepwise for each SOC point of

the battery, the voltage response can be obtained. Figure 8 shows the results for a battery's cell when the DCIR current profile is applied. The battery model parameters in Figure 3 are extracted as shown in Figure 9a–d.

**Table 1.** Specifications used for experimental testing.

| Specifications | | Parameters |
|---|---|---|
| Battery | Module Voltage | 32.4–50.4 V (SLPB55205130H cell – 12S1P) |
| | Nominal Capacity | 11 Ah |
| | Current | Continuous: 55 A Discharge Peak: 110 A Charge Peak: 22 A |
| Supercapacitor | Module Voltage | 18–36 V (ESHSP-1700C0-002R7 – 56S1P) |
| | Capacity | 30.4 F |
| | Current | Rated: 371 A |
| Power Supply for cell | Voltage/Current | 0–60 V, 0–50 A |
| Electronic Load for cell | Voltage/Current | 0–80 V, 0–500 A, Max. 3 kW |
| Power Supply for modules | Voltage/Current | 0–60 V, 0–50 A (DCS60-50E) |
| Electronic Load for modules | Voltage/Current | 0–80 V, 0–500 A, Max. 3 kW (SLL-5K) |

The accuracy of battery modeling using the extracted modeling parameters was verified under pulse current and dynamic current conditions as shown in Figure 10. The upper waveform in Figure 10a shows the current profile, and the SOC estimation result using the ampere-hour (Ah) counting method is shown in the second figure. The capacity measured at room temperature before the experiment was 11.58 Ah. Thus, to discharge the SOC at 10% intervals, a current of 5.5 A was applied for 758 seconds. In the experimental results, it can be seen that the discharge is performed in nine steps from 100% to 10% of SOC. In the third waveform, the solid line represents the measured voltage of the battery. The dash-dotted line shows the battery voltage (denoted as Modeling 1) when the parameters in Figure 9 are configured as the lookup table according to each SOC. The dashed line represents the battery voltage (Modeling 2) that was modeled using the average value of the extracted parameters. The last waveform shows the modeling error for Modeling 1 and 2. It is evident from the experimental results that even when the average model of the extracted battery model is used, the modeling error is smaller than ± 0.03 V. Therefore, parameters of the battery use the average value. The 12-series battery module was modeled using the average value of the battery's cell parameters, and the modeling accuracy of the battery module was determined to comparable to the cell modeling. The modeling accuracy of the battery module is shown in Figure 10b.

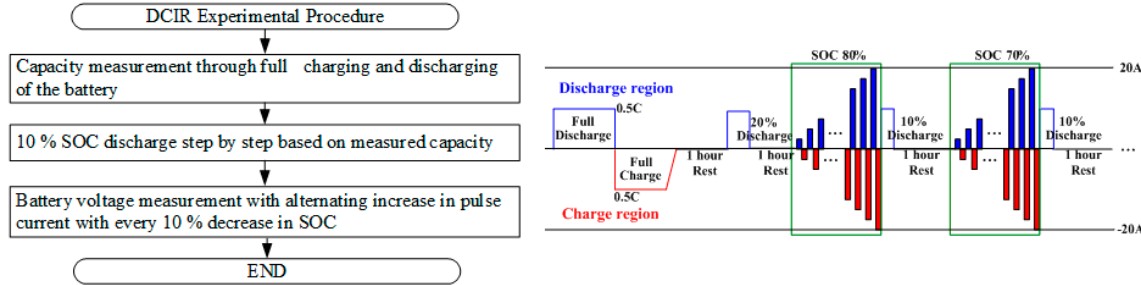

**Figure 7.** Experimental procedure for direct current internal resistance (DCIR) profile.

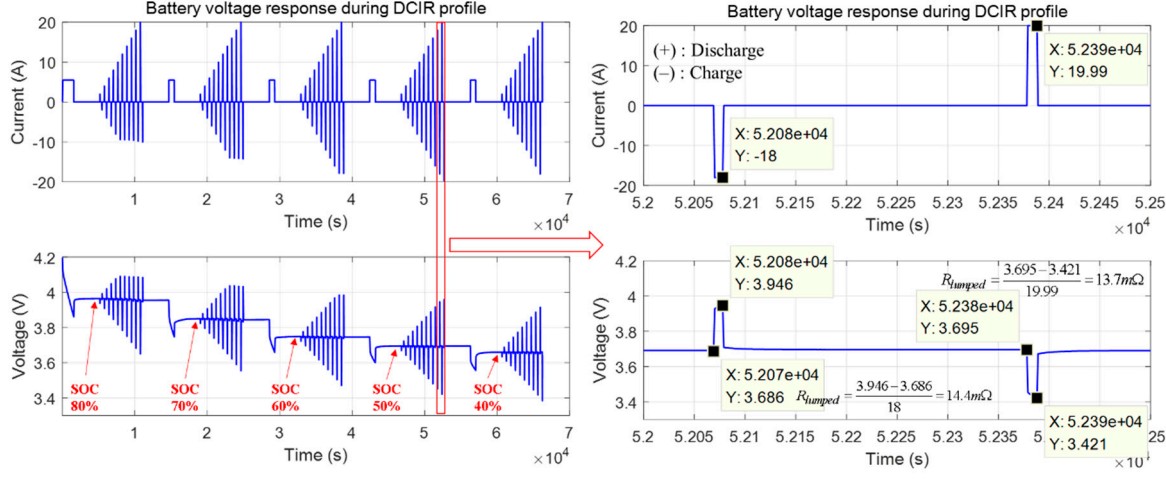

(**a**) Voltage and current responses       (**b**)Enlarged waveform at 50% state of charge (SOC)

**Figure 8.** Current and voltage response of the battery cell under DCIR profile.

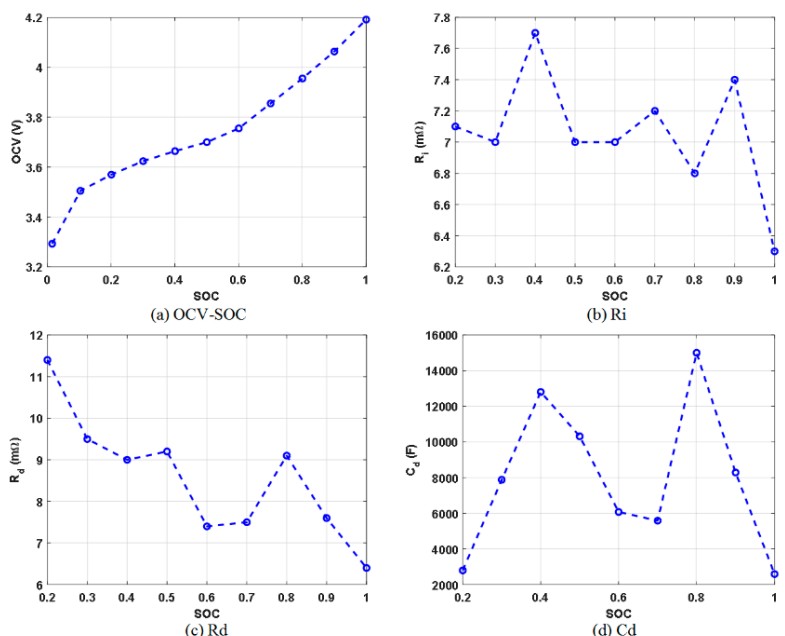

**Figure 9.** Open circuit voltage (OCV) and other parameters of the battery's cell with respect to SOC.

Equations (1)–(5) can be used to calculate the lumped resistance for 10 seconds, and the results of the estimated lumped resistance are shown in Table 2. The lumped resistance of the battery's cell shows a small deviation according to the SOC, but has a value of approximately 14 mΩ. The pulse power according to the SOC of the battery module can be obtained using Equations (10) and (11) as shown in Figure 11. Figure 11a shows the magnitude of the discharging pulse power of the conventional method and the proposed method. The conventional method calculates the pulse power using only the second equation of Equation (10). Therefore, when the pulse current of the battery estimated from Equations (6) to (9) is limited, the pulse power is fixed to a constant value. However, if the battery current is fixed at the maximum value, the pulse current must be calculated from the voltage calculated from Equations (2)–(5). Therefore, the pulse power can be calculated as described in the first Equation of (10), which is larger than the conventional method. On the contrary, in the case of charging, as the state of charge of the battery is lowered, the magnitude of the charging current is fixed to the maximum current from the Equations (6)–(9). At this time, the conventional method calculates the pulse power based on the maximum voltage even if the pulse charging current is limited. However,

the proposed method predicts the magnitude of the pulse power based on the estimated voltage when the pulse current is applied, and this value shows a smaller pulse power value than the conventional method. Figure 11b shows the charge/discharge pulse power capability during the sampling period of 10 seconds and 5 seconds using the proposed method. Because batteries have more energy than supercapacitors, which will be discussed in the next section, there is no significant difference in the amount of pulse power with respect to the sampling period. However, if we want to use the battery for maximum performance in the stable voltage range, we need to calculate the pulse power considering the sampling period of the algorithm.

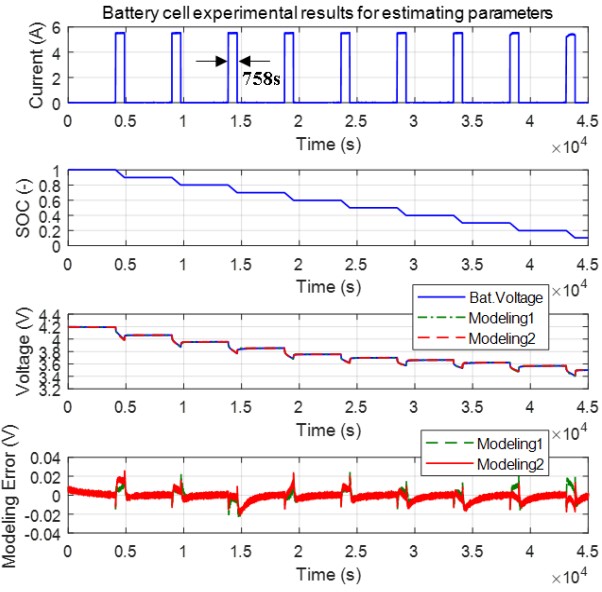

(**a**) Cell modeling verification result

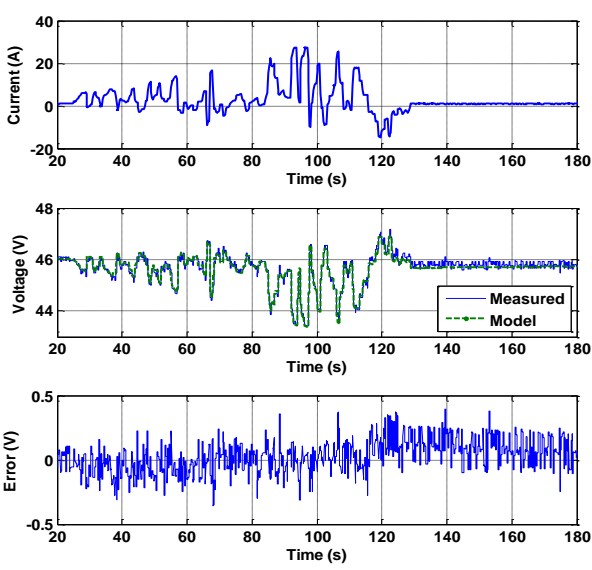

(**b**) Module modeling verification result

**Figure 10.** Modeling verification results of the battery's cell and module.

**Table 2.** Calculated lumped resistance of the battery's cell. Unit: [mΩ].

| SOC | Charging Current | | | | Discharging Current | | | | |
|-----|------|------|------|------|------|------|------|------|------|
| | **4 A** | **8 A** | **12 A** | **16 A** | **4 A** | **8 A** | **12 A** | **16 A** | **20 A** |
| **80%** | 14.2 | 14.5 | | | 14 | 14.9 | 15 | 15.3 | 15.3 |
| **70%** | 14.5 | 14.5 | 14.4 | | 14 | 14 | 14.3 | 14.3 | 14.6 |
| **60%** | 14 | 14 | 14 | 14 | 13 | 13.1 | 13.3 | 13.4 | 13.8 |
| **50%** | 14.2 | 14.2 | 14 | 14.1 | 13 | 13 | 13.2 | 13.4 | 13.7 |
| **40%** | 14.3 | 14.3 | 14.3 | 14.3 | 12.8 | 13 | 13.2 | 13.4 | 13.9 |

The simulation results obtained when the proposed method was applied to the power management algorithm of the electric vehicle are shown in Figure 12. This figure shows the result of charging and discharging the battery with the power scaled down according to the experimental setup, assuming that the ten-ton electric vehicle continuously drives the heavy duty-urban dynamometer driving schedule (HD-UDDS) and the city driving profile [16,17]. The upper waveform in Figure 12a shows the speed profile, and the lower waveform shows the demanded power required from the battery. In this experiment, we conducted a power test on the demanded power profile at 70% of the SOC of the battery, and predicted the magnitude of the pulse power considering the voltage at the point indicated by $V_{bat,dc}$ in Figure 6b as the voltage of the battery. Thus, the magnitude of the pulse power due to the lumped resistance, which includes a 30 mΩ resistor representing the effects of the fuse, the molded case circuit breaker (MCCB) and the cable impedance, was calculated. In this case, the charging pulse power not exceeding the maximum voltage of 50.4 V and the discharging pulse power not exceeding 36V are shown in Figure 12b. The green dashed line represents the discharging pulse power, and the red dash-dotted line shows the charging pulse power. Figure 12c shows the results of a charge/discharge test when the battery is not used within the magnitude of the pulse power estimated by the battery management system (BMS). In Figure 12c, points A and B represent the intervals in which the battery is used in excess of the charging pulse power, and during this period, it can be seen that the battery exceeds the overvoltage range of 50.4 V. On the other hand, Figure 12d shows the simulation result for the case that the charging power is limited to the estimated charging pulse power by the BMS. In this case, since the power of the battery is limited to the available charging power, it can be seen that no overvoltage occurs in the A and B points.

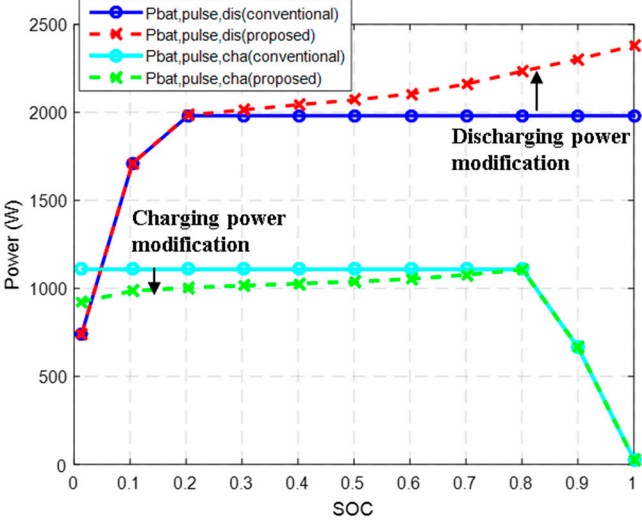

(**a**) Ten-seconds pulse power using the conventional and proposed method

**Figure 11.** *Cont.*

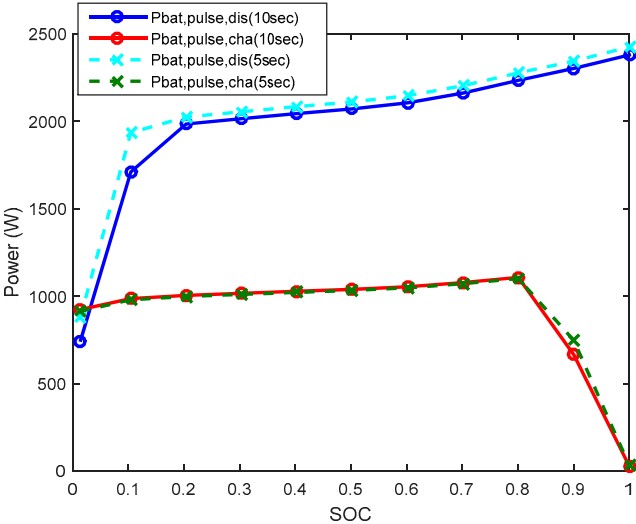

(**b**) Ten- and five-seconds pulse power using the proposed method

**Figure 11.** The proposed and conventional pulse power capability of the 12S1P battery module.

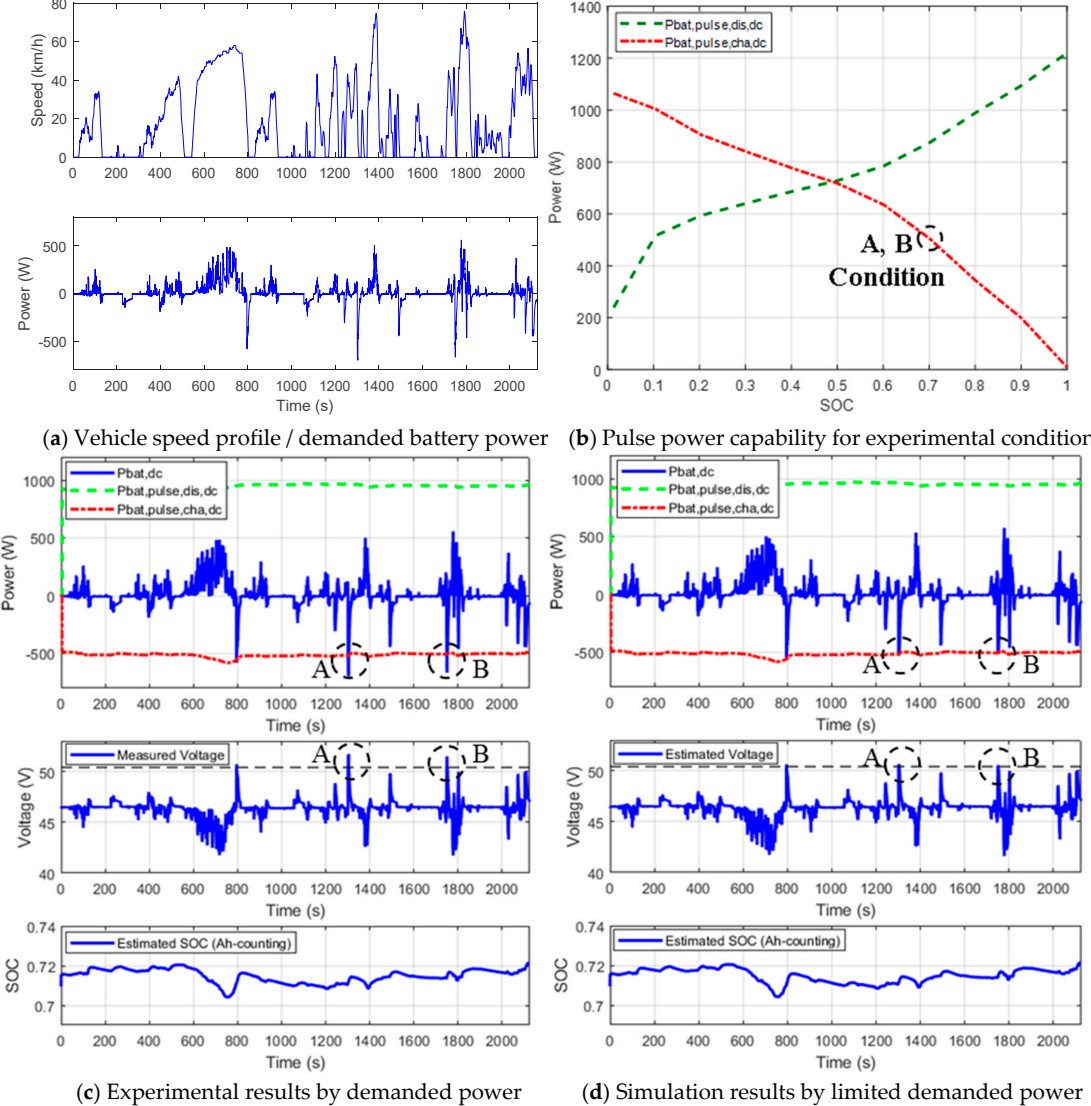

(**a**) Vehicle speed profile / demanded battery power　　(**b**) Pulse power capability for experimental condition

(**c**) Experimental results by demanded power　　　　　(**d**) Simulation results by limited demanded power

**Figure 12.** Battery charging and discharging result by a scaled-down power profile of the electric vehicle.

The parameters of the supercapacitor (SC) can be obtained from the voltage response characteristics for a pulse current, similar to the battery. The series resistance parameter of Figure 5, which is the electrical equivalent circuit of the supercapacitor, can be obtained from the voltage drop of the applied pulse current. The resistance was obtained as shown in Figure 13. The series resistance component of the supercapacitor did not exhibit a significant difference according to the magnitude of the current. An average value of 42.3 mΩ was determined as the series resistance value.

Because of the high power density of super-capacitors, the power capability of the super-capacitors is higher than that of the battery. However, since the energy density is low, the magnitude that can maintain the pulse power for a certain period fluctuates depending on the conditions such as the energy state. Therefore, the pulse power must be estimated considering the sampling period for estimating the pulse power.

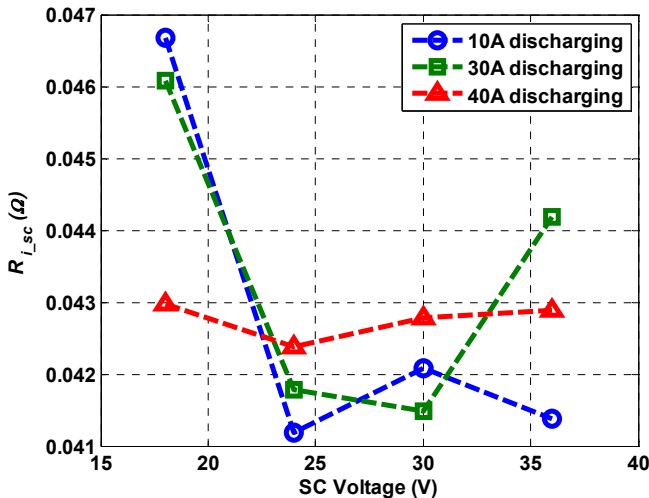

**Figure 13.** Series resistances of the supercapacitor with respect to the voltage and current.

The supercapacitor used in the experiments in this report has a value of 36 Vmax and an equivalent series resistance of 42.3 mΩ. Figure 14 shows the pulse power estimation results when the sampling period was set to 5 seconds or 1 second using Equations (12)–(16). It is evident from the two shaded areas shown in the figure that the pulse output power of the supercapacitor for 5 seconds and 1 second varies depending on the required time.

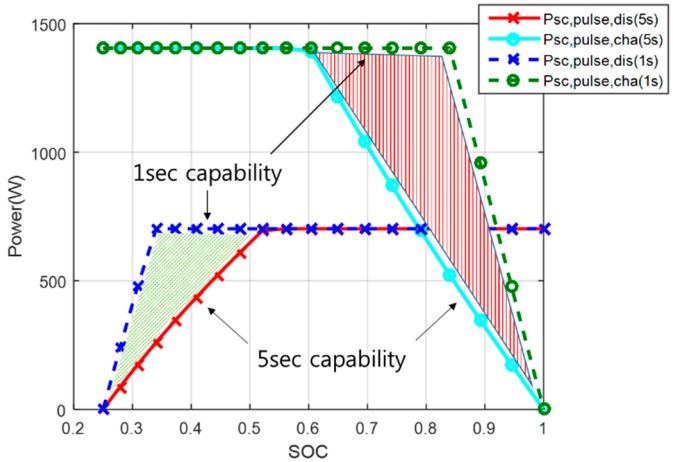

**Figure 14.** Supercapacitor pulse power capability for 1 second and 5 seconds.

As previously indicated, the power distribution algorithm of electric vehicles as shown in Figure 1 uses the power capability of the energy storage device to calculate the torque command reference of the

traction motor. Therefore, it is possible to influence the acceleration performance and the regenerative braking energy of the system, according to the algorithm's sampling period, to calculate the power capability of the supercapacitor. In addition, if the lumped resistance used to calculate the pulse power of the supercapacitor is not calculated based on the sampling period, it is evident that a larger pulse power can be output during discharge. However, it cannot be used, as shown in Figure 14. During the charging process, an issue arises in that it is assumed that the energy storage device can accept an output larger than the actual power capacity.

To predict the pulse power of the supercapacitor, the discharge power capability can be estimated using Equations (17)–(19), considering the characteristics of the cell and the current limitation according to the current specification of the power conversion unit, and the protection relay. Assuming that the pulse power lasts for a certain period (ΔT), the lumped resistance is calculated using the parameters of the supercapacitor. The voltage generated after the duration of the pulse power is then estimated, and the discharge pulse power is estimated using this value.

$$P_{sc,pulse,dis} = V_{sc,pulse,dis} \cdot I_{sc,pulse,dis} \tag{17}$$

$$I_{sc,pulse,dis} = I_{sc,\max} \ , \quad V_{sc,pulse,dis} = V_{sc,t1} \cdot I_{sc,pulse,dis} \cdot R_{sc,lumped} \quad (if \ V_{sc,pulse,dis} > V_{sc,\min}) \tag{18}$$

$$I_{sc,pulse,dis} = \frac{V_{sc,t1} - V_{sc,\min}}{R_{sc,lumped}} \ , \quad V_{sc,pulse,dis} = V_{sc,\min} \quad (if \ V_{sc,pulse,dis} \leq V_{sc,\min}) \tag{19}$$

The charging pulse power capacity of the supercapacitor can be derived using Equations (20)–(22), as in the previous case.

$$P_{sc,pulse,cha} = V_{sc,pulse,cha} \cdot I_{sc,pulse,cha} \tag{20}$$

$$I_{sc,pulse,cha} = -I_{sc,\max} \ , \quad V_{sc,pulse,cha} = V_{sc,t1} - I_{sc,pulse,cha} \cdot R_{sc,lumped} \quad (if \ V_{sc,pulse,cha} < V_{sc,\max}) \tag{21}$$

$$I_{sc,pulse,cha} = \frac{V_{sc,t1} - V_{sc,\max}}{R_{sc,lumped}} \ , \quad V_{sc,pulse,cha} = V_{sc,\max} (if \ V_{sc,pulse,cha} \geq V_{sc,\max}) \tag{22}$$

As previously indicated, the pulse power capability for the case where the sampling period of the pulse power of the supercapacitor is set to 1 second is shown in Figure 15, for comparison of the conventional method and the proposed method. In this case, $P_{sc\_pulse,dis}$(*conventional*) and $P_{sc\_pulse,cha}$(*conventional*) represent the discharging and charging pulse power capability for 1 second for the conventional method. The power parameters with 'proposed' in the notation represent the charging and discharging pulse power capability for the proposed method. As the SOC of the supercapacitor increases in the discharge period, more power can be supplied using the proposed method compared to the conventional method. Moreover, in the charging period, the proposed method has a lower charge at a smaller SOC compared to the conventional method. Therefore, when the proposed method is applied to the power distribution algorithm of an electric vehicle, the output capability is increased during acceleration, and the performance and stability of the system can be increased by restricting the chargeable power during regenerative braking.

The pulse current prediction results of the supercapacitor are verified in Figure 16. After the supercapacitor was charged to 36 V, an experiment was conducted to facilitate discharge to the lowest possible voltage with a pulse current of 39 A within a specified period. The controller unit was used to calculate the pulse power of the supercapacitor with a sampling period of 1 second and serves to limit the current command when a value that exceeds the available pulse power is applied. In Figure 16a,

since a current command of 39 A is greater than the pulse power that can be discharged at 43 seconds, it is evident that the magnitude of the pulse current changed to 29 A. This value is the predicted one-second pulse current. Since the voltage of the supercapacitor after the one-second pulse current of 29 A was applied became 18 V, which is a low voltage setting value, it can be seen that the magnitude of the estimated one-second pulse current is accurate. The top waveform of Figure 16b shows the magnitude of the power of the supercapacitor during the experiment, and the bottom waveform shows the powers based on the state of charge of the supercapacitor. The red dashed line indicates the magnitude of the discharging pulse power of Figure 15, and it can be seen that the pulse power estimation result and the experimental result are almost the same.

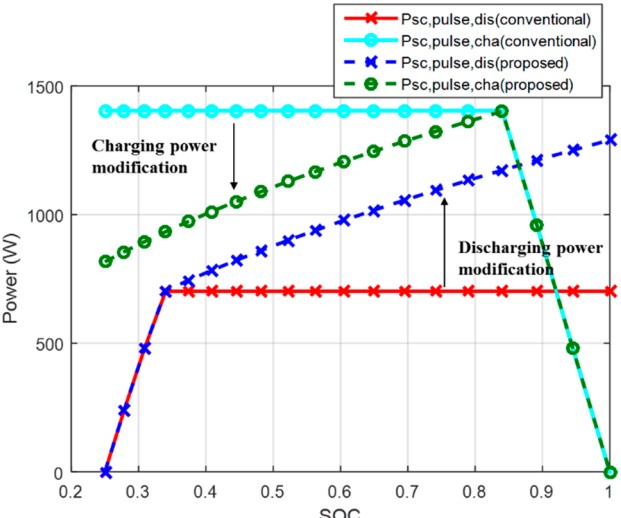

**Figure 15.** The pulse power capability of the supercapacitor for 1 second using the proposed method.

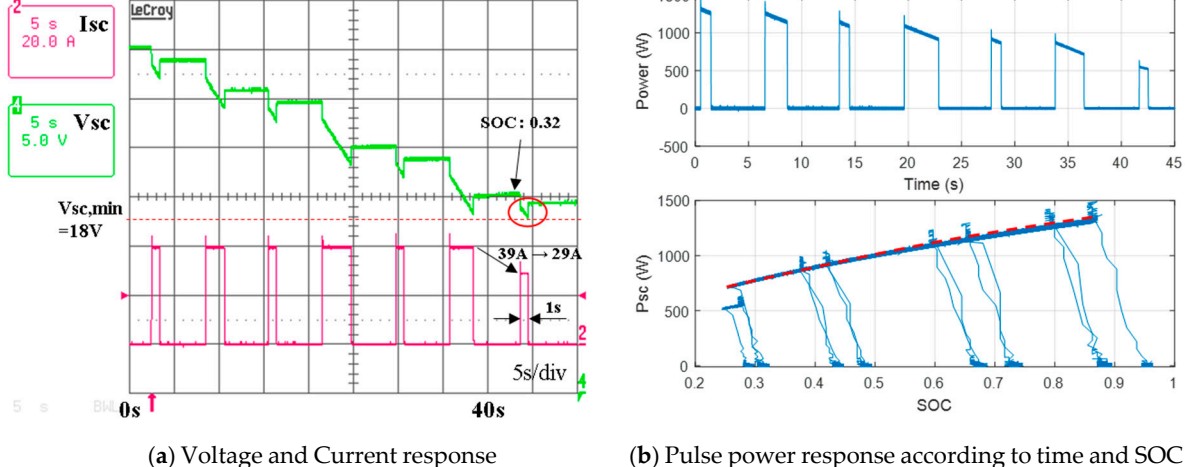

(**a**) Voltage and Current response        (**b**) Pulse power response according to time and SOC

**Figure 16.** Pulse power estimation result of supercapacitor for verification of the proposed method.

## 5. Conclusions

In this report, we propose a method for estimating the pulse power capability of batteries and supercapacitors. The proposed method estimates the lumped resistance and the pulse current after estimating the voltage after the sampling period at which the pulse power is calculated from the equivalent electrical circuit modeling. Then, the magnitude of the pulse power is predicted according to whether or not the magnitude of the estimated current is the maximum value. The detailed analysis and experimental results are presented to evaluate the validity of this study based on the observed

differences between the proposed method and the conventional method according to pulse duration. The results of this study can be extended to analyze the equivalent model of a battery with series resistance and two or more RC ladders. If the parameters of the battery and supercapacitor vary with temperature, additional studies should be applied. If only the results of this study are applied to the algorithm, the effect of parameter variations can be minimized by applying a lookup table according to temperature. To demonstrate the validity of the proposed method, pulse power experimental results for a lithium battery module with 44.4 V, 11 Ah, and a supercapacitor with 36 V, 30 F were presented. The pulse power proposed in this paper can be applied online or via a look-up table method in a supervisory controller of an electric vehicle or an energy storage system to achieve optimal power distribution. In addition, the proposed method can increase the performance and stability of storage system when applied to the system control logic in combination with the estimation of the SOC of the battery.

**Author Contributions:** S.L. contributed to the main idea of this article, and wrote the paper. S.L. and J.K. performed the experiments. J.K. revised the paper critically. All authors approved the final version to be published.

**Acknowledgments:** This study was supported by research fund from Chosun University, 2018 (K207462002-1).

**Conflicts of Interest:** The authors declare no conflict of interest.

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
