# Peer review of "Power Capability Analysis of Lithium Battery and Supercapacitor by Pulse Duration"

_electronics, doi:10.3390/electronics8121395_

Round 1

Reviewer 1 Report

The Authors visibly improved quality of the paper and they taken into account all my suggestions.

In my opinion this paper could be published in its current form.

Reviewer 2 Report

Thank you for the revision.

This manuscript is a resubmission of an earlier submission. The following is a list of the peer review reports and author responses from that submission.

Round 1

Reviewer 1 Report

The paper tried to handle an interesting topic, which is the energy management for battery-supercapacitor hybrid energy storage system, however, the some improvements are required. The below are my comments:

1- Although the authors claimed the accurate estimation of the pulsed power capability of the battery according to the estimated lumped resistance, the used model for the battery lumped resistance presented in the paper is inaccurate due to the utilization of single R//C branch (single time constant) in series with the Ohmic resistance, whilst the simplest model that can give reasonable accuracy should has at least two time constants (R//Cs) in series with the Ohmic resistance: one describing the charge transfer process within the electrolyte and the other for describing diffusion process within the electrodes. Of-course, some simplification can be applied according to the time duration to be considered, but this should be clearly described and the duration to be considered should be clearly stated as it cannot be generalized. 

2- The paper not considered the change of the parameters of the battery model due to the change in temperature which is more significant compared to its changes due to battery state of charge. Temperature also should may need to be taken into account when deciding the power capability of the lithium-ion battery.  

3-The authors used the term '' sample time'' many times within the paper to describe the pulse duration (for example at line 77 and line 125), this may cause confusions, as the term sample time is normally used to describe the time for sampling the measured variables that should be significantly less than the pulse duration in order to acquire reasonable number of measurements that help in predicting the estimates for model parameters or processing of the measured values; i.e filter the noise,….etc.

4- Equation (3) describes the decrease in the battery state of charge (in Ah), and Not the decrease in the battery energy as stated in line 87, as the battery voltage is not considered in this equation.

5- It is not clear how equation (14) was derived from equation (13).

6- In the experimental setup shown in Figure 6 (a), the device named ‘’EIS’’ and its function was not descried in the text.

7- Figure 8 should include a zoom in for one cycle for the reader to be able to see I/V relationship that can allow the observation of the DC resistance from the plot.

8- Figure 10 (a) seems to has a problem, according to the figure, there are Nine consecutive discharging pulses each with a current of ≈5.5A for a duration of ≈1000s. This results in a total discharge of 13.75 Ah which is more than the specified capacity of the battery cell (11Ah) that mentioned in Table1. This result need for more clarifications and a plot for the battery state of charge during this experiment is need to be included.

9- In line 216, it is mentioned that the power capability of the ‘’source’’, but it is not clear which source, it seems to be a typo and the authors was trying to say the power capability of the ‘’supercapacitor module’’

Reviewer 2 Report

The quality of the paper is quite low, and it is not organized well. Authors just show results without providing sound explanation. The efficacy of the proposed method is not well demonstrated. The benefits of the proposed method is not well explained. Authors must check and read your own paper before submission.

Line 13

ESS is not defined. This term is defined in line 275 for the first time.

AP and BP in Fig. 1 are not defined. Authors must check the paper carefully before submission.

Line 184-188

Lack of explanation. Authors just write about the figure but failed to explain the reason or mechanism behind this figure. Why does the proposed method allow hither power capability at high SOC for discharging? Why is the power capability of the proposed method inferior at low SOC? Why can both method not supply high power at high SOC? This paper totally lacks explanations.

12

What do you want to demonstrate in Fig. 12? What was demonstrated in this figure? The proposed method is nether demonstrated nor compared with the conventional one in this figure.

12

Why do you intentionally cover the periods A-C? Authors discuss these periods while covering the results. Readers would have no idea what is happening in these periods.

The definition of sampling time is very unclear. Do you use the term 'sampling interval' to represent the pulse duration?

Line 249

No sound explanation presented. You just describe the results without providing the sound reason.

16

The right-hand figure is no mentioned in this paper.

Reviewer 3 Report

The reviewed paper is devoted to analysis of power capability of lithium battery and supercapacitors at operation with pulsed current. The Authors proposed equivalent circuit of the considered devices, elaborated equations describing properties of these devices and design set-ups for measurements their properties. Some results of calculations and measurements were shown. 

The paper includes interesting results, but the presentation of the obtained results is difficult to understand. The Authors should improve considered paper before publishing. In the revised version the aim of the paper should be clearly stated and it should be written, why this aim is important. The Conclusions should be extended and describe most important results obtained by the Authors. In some figures it is not visible, what is presented in them - results of calculations or results of measurements. The form of some equations is more adequate to computer program than for people - this should be changed.